# Novel role of SARM1 mediated axonal degeneration in the pathogenesis of rabies

**Vinod Sundaramoorthy**[1]*, **Diane Green**[1], **Kelly Locke**[1], **Carmel M. O'Brien**[2,3],
**Megan Dearnley**[1], **John Bingham**[1]

1 CSIRO Australian Animal Health Laboratory, East Geelong, Victoria, Australia, 2 CSIRO Manufacturing,
Research Way, Clayton, Victoria, Australia, 3 Australian Regenerative Medicine Institute, Monash University,
Clayton, Victoria, Australia

* vinod.sundaramoorthy@csiro.au

## Abstract

Neurotropic viral infections continue to pose a serious threat to human and animal wellbeing.
Host responses combatting the invading virus in these infections often cause irreversible
damage to the nervous system, resulting in poor prognosis. Rabies is the most lethal neuro-
tropic virus, which specifically infects neurons and spreads through the host nervous system
by retrograde axonal transport. The key pathogenic mechanisms associated with rabies
infection and axonal transmission in neurons remains unclear. Here we studied the patho-
genesis of different field isolates of lyssavirus including rabies using ex-vivo model systems
generated with mouse primary neurons derived from the peripheral and central nervous sys-
tems. In this study, we show that neurons activate selective and compartmentalized degen-
eration of their axons and dendrites in response to infection with different field strains of
lyssavirus. We further show that this axonal degeneration is mediated by the loss of NAD
and calpain-mediated digestion of key structural proteins such as MAP2 and neurofilament.
We then analysed the role of SARM1 gene in rabies infection, which has been shown to
mediate axonal self-destruction during injury. We show that SARM1 is required for the
accelerated execution of rabies induced axonal degeneration and the deletion of SARM1
gene significantly delayed axonal degeneration in rabies infected neurons. Using a microflui-
dic-based ex-vivo neuronal model, we show that SARM1-mediated axonal degeneration
impedes the spread of rabies virus among interconnected neurons. However, this neuronal
defense mechanism also results in the pathological loss of axons and dendrites. This study
therefore identifies a potential host-directed mechanism behind neurological dysfunction in
rabies infection. This study also implicates a novel role of SARM1 mediated axonal degener-
ation in neurotropic viral infection.

## Author summary

Lyssaviruses including rabies, still causes devastating loss of human life every year and
many victims are children under the age of 15. Rabies infection causes severe neurological
dysfunction in the host resulting in paralysis, cognitive deficits and behavioural abnormal-
ities. The mechanism of how rabies infection induces neurological dysfunction in the host

ppat.1008343

Jefferson University, UNITED STATES

**Data Availability Statement:** All relevant data are
within the manuscript and its Supporting
Information files.

**Funding:** The authors received no specific funding for this work.

**Competing interests:** The authors have declared that no competing interests exist.

remains unclear. This is because unlike other microbial infections, rabies infection rarely causes neuronal cell death and loss of neurons in the host nervous system. In this study, we show that neurons activate specific axonal self-destruction mechanism during rabies infection to prevent the spread of virus. However, this neuronal self-defense mechanism results in the loss of axons and dendrites, the structural components essential for the functioning of neurons. We further show that axonal degeneration in rabies infection is mediated by SARM1 gene, which has been previously shown to mediate defensive self-destruction of axons and dendrites in the event of neuronal injury. In summary, this study identifies a novel molecular mechanism behind neuronal dysfunction in rabies infection. This study also describes a novel intrinsic anti-viral defence mechanism in neurons, which could influence the pathogenesis of neurotropic viral infections.

## Introduction

Neurons have an intrinsic molecular mechanism to self-destruct their axons and dendrites in response to injury or stress. This self-destruction serves as a defensive mechanism to remove injured or unhealthy neurites preventing further damage to the nervous system. However an aberrant or over-activation of this mechanism could result in the dysfunction of the nervous system and has been implicated in many neurodegenerative diseases [1, 2]. Primarily, axonal self-destruction is linked with the levels of an essential energy molecule, nicotinamide adenine dinucleotide (NAD), in the axons [2]. NAD loss corresponds with axonal degeneration and the overexpression of NMANT2, a NAD biosynthetic enzyme that has been shown to significantly delay axonal degeneration [3]. More importantly, Sterile Alpha and TIR Motif-containing 1 (SARM1) was recently found to be a key facilitator of axonal degeneration because of its direct ability to digest NAD [4, 5]. Current models suggest that when NAD levels in the axon fall below a critical threshold, a suicide signaling cascade is triggered by the accelerated digestion and loss of residual NAD by SARM1. Depletion of NAD triggers an influx of calcium ions into the axons, which then activates several calcium dependent calpains. These calpains in turn regulate the proteolytic digestion of dendritic and axonal cytoskeletal proteins such as microtubule associated protein 2 (MAP2) and neurofilament [6–10], resulting in the degeneration of both dendrites and axons. Morphologically this calpain-mediated axonal degeneration is characterized by the appearance of focal axonal swellings, followed by granular disintegration of axons [11]. While terms such as axonal degeneration or axonal self-destruction are broadly used to refer to this SARM1-mediated pathway, this mechanism could result in the degeneration of dendrites in addition to axons, causing an extensive pathology of neurites.

Axonal self-destruction mechanism has been associated with neuronal trauma or stress, but its role in neurotropic viral infections remains unknown. It is particularly intriguing to examine the possible activation and consequence of axonal self-destruction during infection with viruses such as rabies, West Nile virus, Herpes simplex virus (HSV) and polio virus [12, 13], where axonal transmission of virus is involved.

Rabies virus (Family Rhabdoviridae, genus Lyssavirus) is the most lethal among viruses which invade and infect the nervous system. Lyssaviruses infect the peripheral nerve endings and travel exclusively within the axons of the peripheral and central nervous system causing fatal encephalitis [14]. In humans, the majority of rabies infections (about 80%) are classified as the furious form, characterized by agitation, aggression, hallucination, hydrophobia and confusion. In animals, symptoms include aggression, disorientation, loss of fear and the urge to bite. Following the acute excitatory phase, the host lapses into coma and eventually death

occurs due to cardio-respiratory arrest. In some cases (about 20%), rabies infection causes muscle weakness and ascending paralysis without the excitatory phase, resulting in the paralytic form of rabies, which is also almost invariably fatal [15–17]. In spite of the severe neurological signs, pathological lesions typical of other viral encephalitis such as gliosis, perivascular lymphocytic infiltration, phagocytosis of neurons or neuronal apoptosis are rarely observed in rabies patients during the acute disease [18–20]. However structural damage to neurites such as focal axonal swelling, segmental demyelination and impairment of synaptic transmission are observed in natural [21–25] and experimental rabies infections [26–31]. This evidence suggests that rabies infection could induce pathological damage to the axons and dendrites of neurons without affecting the neuronal cell body. Such a compartmentalized degeneration of neurites could underlie neurological dysfunction in rabies infection. However, the basic molecular mechanism of how the rabies infection causes selective neurite damage remains unknown.

## Results

### Rabies infection induces selective degeneration of dendrites and axons in mouse neurons derived from the central and peripheral nervous system

This study examines the neuronal pathogenesis of different field strain lyssaviruses originating from infected bats or dogs, which are subjected to minimal laboratory-adaptation. Unlike tissue culture adapted or attenuated rabies strains, these field isolates are expected to retain original pathogenicity [32–34]. Dog strain rabies virus isolated from domestic dogs from Zimbabwe (Z.Dog) and Thailand (T.Dog), and two bat lyssaviruses from a silver-haired bat (*Lasionycteris noctivagans*) from Canada (SHBRV) and an Australian bat Lyssavirus isolated from an infected horse (H.ABLV) are compared with mock infection (treatment with tissue culture supernatant from uninfected Neuro-2a cells used for viral stock amplification) or an uninfected control group. Infection of primary cortical neurons with all the strains at MOI of 1, induced focal swellings in the neurites of cortical neurons at 16 hours post infection (Fig 1A, S1A Fig). At 24 hours of infection, we observed complete loss of dendritic MAP2 staining (Fig 1B and 1C, S1B Fig). Specific degradation of key axon and dendritic structural protein was further observed by western blotting of neuronal lysates infected with lyssavirus strains. Western blotting showed that infection with lyssavirus resulted in significant degradation of MAP2 and neurofilament proteins (Fig 1D and 1E). To further confirm the loss of axons and dendrites in field strain rabies infection, we infected cortical neurons with parental viral stocks directly isolated from infected dog (T.Dog strain, P0) or after first passaging in sucking mouse brain (Z. Dog strain, P1). This infection resulted in distinct loss of both MAP2-positive dendrites and neurofilament-positive axons at 24 hours post infection (S2A Fig). While the structural proteins in the neurites were specifically degraded, the cell bodies, including the nucleus, were observed to be unaffected (Fig 1B, S1B Fig). This specific damage to neurites was also observed by live cell imaging of cortical neurons, which displayed swelling and granular disintegration of neurites within a timeframe of 20 hours post-infection with SHBRV lyssavirus (S1 Video). To examine whether the loss of neurites in rabies infection is due to the induction of apoptosis, TUNEL (Terminal deoxynucleotidyl transferase dUTP nick end labelling) staining was performed (S2B Fig, S2C Fig). This analysis showed no significant difference in the induction of apoptosis in neurons infected with all the lyssavirus strains compared to mock infection, despite the dramatic loss of neurites at 24 hours post-infection (S2C Fig).

Axonal degeneration in primary cortical neurons was then examined in microfluidic chambers, which allows the cell body and axons of neurons to be cultured in physically separate compartments [28] (S3A Fig, S3B Fig). This system enables the localized infection of axons with rabies virus and investigation of pathological events following axonal uptake and

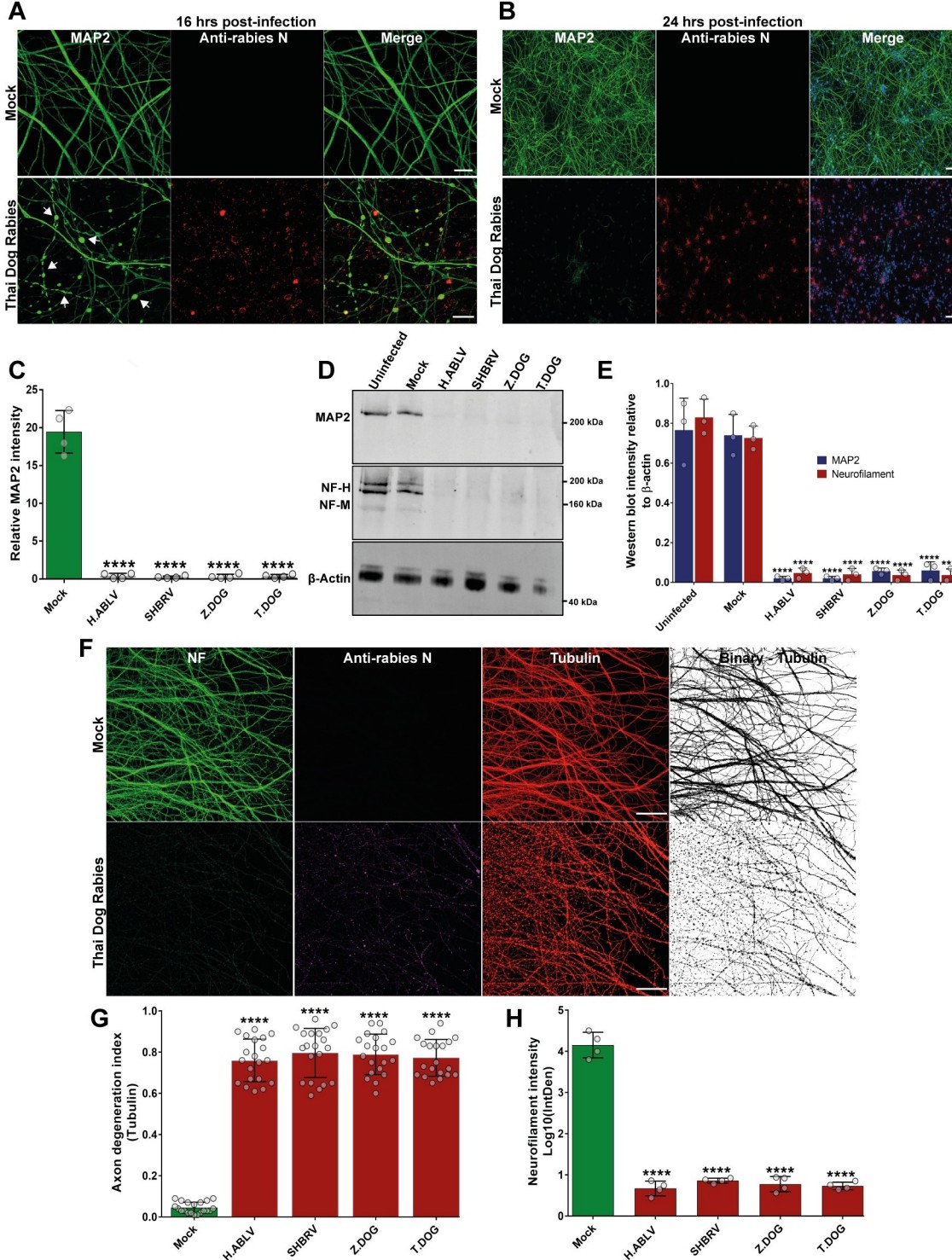

**Fig 1. Rabies infection induces selective degeneration of axons and dendrites in primary cortical and DRG neurons. (A)** Rabies induced blebbing of MAP2-positive neurites (white arrows) in cortical neurons at 16 hours post infection (high magnification). Neurons are stained with anti-MAP2 antibody (green) and anti-rabies nucleoprotein antibody (red). Scale bar 10 μM. Images from other rabies strain infections are shown in S1A Fig. **(B)** Rabies induced loss of MAP2-positive neurites in cortical neurons at 24 hours post infection (low magnification). Fig 1B shows representative stitched tile images of mouse cortical neurons infected with Thai dog rabies. Scale bar 100 μM (images are maximum intensity projection of Z-stacks). Neurons are stained with anti-MAP2 antibody (green), anti-rabies nucleoprotein antibody (red) and DAPI (blue). Images from other rabies strain infections are shown in S1B Fig.

**(C)** Quantification of the loss of MAP2-positive neurites relative to cell nuclei in rabies infection. At least 500 neurons were quantified per sample. ****$p < 0.0001$ versus mock infection, n = 4. **(D)** Loss of axonal (neurofilament) and dendritic (MAP2) proteins in cortical neurons infected with rabies for 24 hours. Representative western blot images of cortical neuron lysates probed with anti-MAP2 antibody and pan-axonal neurofilament antibody. β-actin is shown as the loading control. **(E)** Quantification of MAP2 and neurofilament (heavy and medium chain) intensity in western blots relative to β-actin (loading control). ****$p < 0.0001$ versus controls, n = 3. **(F)** Mouse DRG neurons cultured in microfluidic devices were infected with different street strain rabies virus in the axon panel for 24 hours. Fig 1F shows axonal degeneration in DRG neurons infected with Thai dog rabies, compared with mock infection. The neurons were stained with pan axonal neurofilament antibody (NF, green), anti-rabies nucleoprotein antibody (magenta) and anti α-tubulin (red). Scale bar 100 μM (maximum intensity projection of Z-stacks). **(G)** Quantification of axonal degeneration as a ratio of disintegrated axons versus healthy (filamentous) (Axon degeneration index). ****$p < 0.0001$ versus mock infection, n = 4 (5 representative images quantified from each repeat). **(H)** Quantification of neurofilament immunostaining intensity in the axon panel images shown in Fig 1F & S4A Fig. ****$p < 0.0001$ versus mock infection, n = 4.

retrograde transport of rabies virus, as seen in physiological infection. Axonal infection of cortical neurons with rabies virus (Z.Dog strain) at MOI of 1 for 24 hours resulted in granular disintegration of neuronal axons as observed by tubulin staining (S3C Fig, S3D Fig). Furthermore, specific loss of neurofilament staining in the distal axons was observed at 24 hours post-infection, while the neurofilament staining could still be observed in the neuronal cell bodies and proximal axons (S3C Fig). This reaffirms that axonal loss in rabies infection is a locally executed mechanism independent of the neuronal cell body.

Next, embryonic mouse dorsal root ganglion (DRG) neurons cultured in microfluidic chambers were used to examine whether rabies infection could induce axonal degeneration in neurons derived from the peripheral nervous system. Like the cortical neurons, the DRG neurons infected at the axons with all the strains of lyssavirus displayed granular disintegration of axons with tubulin staining and significant loss of neurofilament staining at 24 hours post-infection (Fig 1F, 1G and 1H and S4A Fig). In addition, live DIC imaging showed selective degeneration of neurites in DRG neurons infected with SHBRV (S2 Video). These data show that infection with rabies virus induces selective degeneration and loss of axons and dendrites in neurons derived from the central and peripheral nervous systems.

## Rabies-induced neurite degeneration is partially inhibited by the addition of NAD and inhibition of calpain activation

The morphological features of axonal and dendritic degeneration induced by rabies infection closely resembled calpain-mediated axonal degeneration (Fig 1, S1 Fig, S2 Fig, S3 Fig). Hence, it was next examined whether rabies induced degeneration involves NAD loss and calpain activation. A panel of cortical neuron cultures were generated, where infected neurons were treated with increasing concentrations of NAD to replace its loss during degeneration. Infected neurons were also treated with EGTA to block calcium influx [35] and calpain inhibitor-III, preventing calpain activation [8]. These studies showed that addition of NAD partially prevented the loss of MAP2-positive dendrites in rabies infection in a dose dependent manner (Fig 2A and 2C, S4B Fig, 24 hours post-infection). In addition, treatment with EGTA or calpain inhibitor-III resulted in significantly higher MAP2 staining compared to untreated controls (Fig 2A and 2C, S4B Fig). However, these infected neurites still displayed significant fragmentation and focal swellings (Fig 2B), suggesting that EGTA and caplain inhibitor-III were only effective in blocking the downstream calpain-mediated proteolytic digestion during axonal degeneration.

## SARM1 is required for the rapid execution of rabies induced axonal degeneration

As SARM1 is a chief facilitator of axonal degeneration by the accelerated digestion of NAD, it was then tested whether SARM1 is required for rabies induced axonal degeneration. Primary

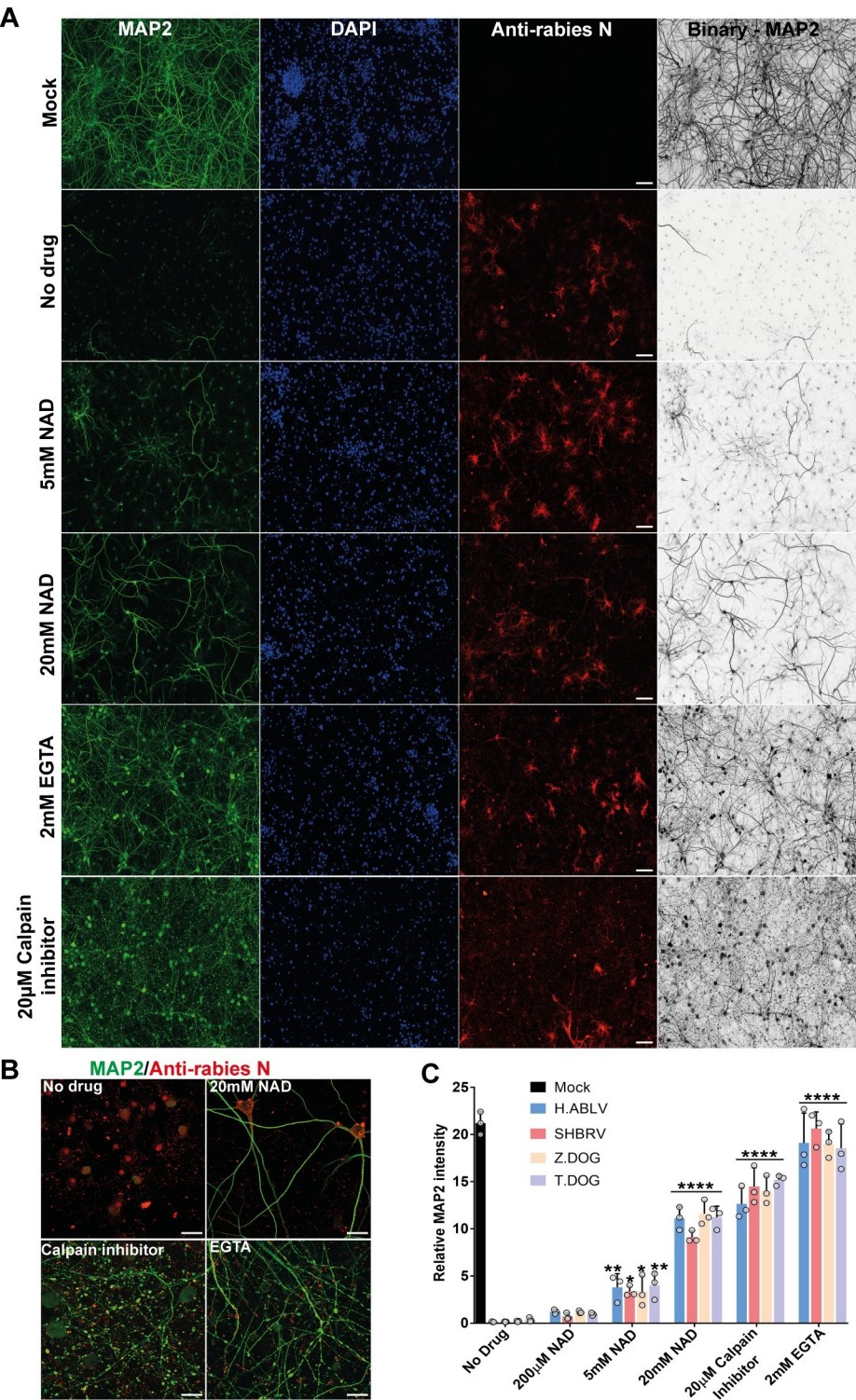

**Fig 2. Pharmacological inhibition of neurite degeneration in rabies infection.** (A) Effect of extracellular NAD addition and calpain inhibition on rabies induced loss of MAP2-positive neurites. Stitched tile images of mouse cortical neurons infected with Thai dog rabies for 24 hours and treated with NAD, EGTA or Calpain inhibitor III. Scale bar 100 µM (maximum intensity projection of Z-stacks). Neurons are stained with anti-MAP2 antibody (green), DAPI (blue) and anti-Rabies nucleoprotein antibody (red). (B) Higher magnification images of MAP2-positive neurites infected with rabies and treated with drugs. Images show significant blebbing in the neurites treated with calpain inhibitor III and EGTA. Scale bar 20 µM. (C) Quantification of MAP2-positive neurites relative to cell nuclei. At least 500 neurons were quantified per sample. ****p < 0.0001 versus mock infection, n = 4.

cortical neurons from SARM1 knockout mice [36] and wildtype mice were infected with lyssa-virus strains. At 24 hours of infection, the wildtype neurons displayed complete loss of MAP2-positive neurites. The SARM1 knockout neurons, however, displayed only minimal loss of MAP2 staining, with intact neurites in a majority of the neurons (Fig 3A). But when the rabies infected SARM1 knockout neurons were cultured over several days, a gradual loss of MAP2 staining was observed (Fig 3A and 3B, S5A Fig). These data suggest that the deletion of SARM1 gene was able to significantly delay loss of neurites induced by rabies infection from 24 hours to beyond 7 days (Fig 3B). Western blotting was then performed on the lysates of wildtype and SARM1 knockout cortical neurons infected with rabies strains for 48 hours. These blots showed significantly higher levels of dendritic MAP2 and axonal neurofilament proteins in rabies infected SARM1 knockout neurons in comparison to wildtype neurons (Fig 3C and 3D). To specifically analyse the degeneration of axons, SARM1 knockout cortical neurons were then cultured in microfluidic chambers and infected with rabies virus (Z.Dog strain) in the axon panel. These neurons had intact axons without any granular disintegration at 24 hours post-infection (S5B Fig). This was in stark contrast to the infected wildtype cortical neurons (S3C Fig). Next primary DRG neurons from the SARM1 knockout mice were cultured in microfluidic chambers and infected with multiple rabies strains in the axon panel. In this study, significantly higher neurofilament staining and reduced axonal degeneration index were observed in the SARM1 knockout DRG neurons as compared to wildtype DRG neurons infected with all the strains of rabies virus used in this study (Fig 3E, 3F and 3G, S5C Fig). Hence, these results show that SARM1 is required for the rapid facilitation of axonal degeneration in response to rabies infection, both in cortical and DRG neurons.

## Axonal degeneration impedes the transmission of rabies virus between neurons

Since we observed SARM1/NAD mediated axonal degeneration in neurons infected with lyssavirus, we hypothesized that this mechanism could be an intrinsic neuronal defense against neurotropic viral infections to block the spread of virus through synaptically-connected neurons. To examine this possibility for lyssavirus, a microfluidic system was used to generate an ex-vivo model of axonally-connected neuronal network. In this model, primary cortical neurons were cultured on neighboring panels, which were connected by the axons projecting through the adjoining microchannels. Neurons in the inoculated panel were infected with rabies virus and the spread of virus through the network of axons to the non-inoculated panel were examined. These neuronal cultures were stained with MAP2 antibody to visualize the dendrites and neuronal cell bodies in each panel. Wildtype cortical neurons infected with SHBRV lyssavirus at MOI of 1 for 24 hours showed extensive degeneration of neurites in the inoculated panel as observed by the loss of MAP2 staining (Fig 4A). This correlated with restriction of viral infection to the inoculated panel with limited spread of virus to the neurons in the non-inoculated panel (Fig 4A). However, when the same experiment was repeated with SARM1 knockout neurons, the SHBRV lyssavirus was able to efficiently spread from inoculated to non-inoculated panel. This was associated with intact MAP2-positive dendrites in the inoculated panel, unlike the wildtype neurons (Fig 4B). To quantify the amount of infectious virions in the panels, the culture media from both the inoculated and non-inoculated panels were collected for titration assays (Fig 4C) and the viral titre ratio between the panels were compared (Fig 4D). No significant difference in viral titre was observed in the culture media collected from inoculated panels of wildtype and SARM1 knockout neurons (Fig 4C). But the viral titre was significantly reduced in the non-inoculated panel of wildtype neurons compared to SARM1 knockout neurons (Fig 4C). Further analysis also showed a significantly higher titre

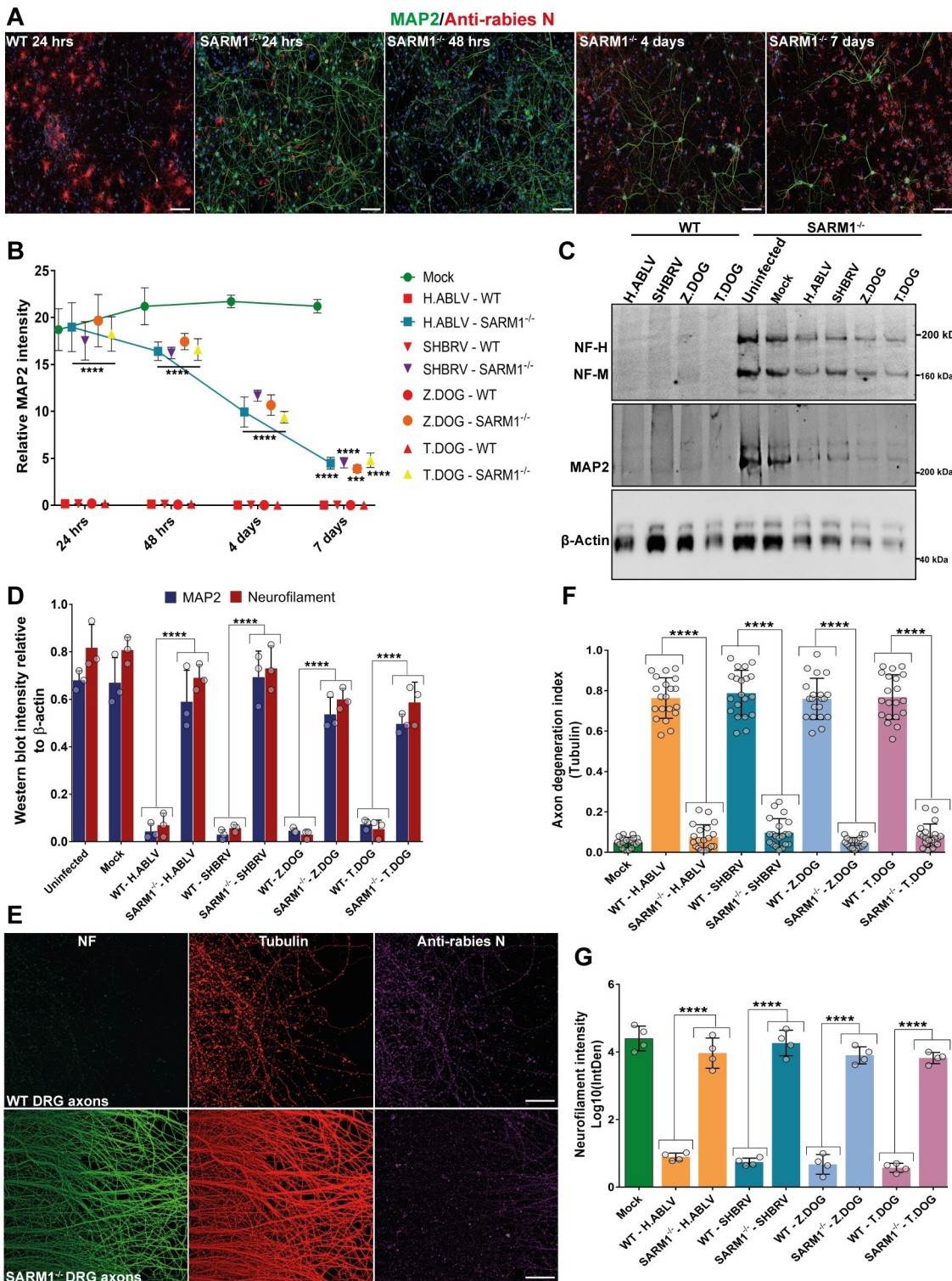

**Fig 3. Deletion of SARM1 gene significantly delays axonal degeneration in primary cortical and DRG neurons infected with rabies. (A)** Time course analysis of neurite degeneration in SARM1 knockout neurons infected with rabies. Representative stitched tile images of mouse cortical neurons from wildtype (WT) and SARM1 knockout (SARM1$^{-/-}$) mice infected with Thai dog rabies for 24 hours to 7 days. Images show intact MAP2-positive neurites in rabies infected SARM1$^{-/-}$ neurons at 24 hours, but a gradual loss of MAP2-positive neurites extending up to 7 days. Neurons are stained with anti-MAP2 antibody (green), anti-rabies nucleoprotein antibody (red) and DAPI (blue). Scale bar 100 μM (maximum intensity projection of Z-stacks). **(B)** Quantification of MAP2-positive

neurites relative to cell nuclei in WT, SARM1$^{-/-}$ cortical neurons infected with different strains of rabies virus from 24 hours up to 7 days. MAP2 intensity is quantified relative to the number of DAPI stained nuclei. At least 500 neurons were quantified per sample. ****p < 0.0001, ***p < 0.001 versus mock infection, n = 3. **(C)** Western blotting of MAP2 (dendritic) and neurofilament (axonal) proteins in cortical neurons from WT and SARM1$^{-/-}$ mice infected with different rabies strains for 48 hours. β-actin is shown as the loading control. **(D)** Quantification of MAP2 and neurofilament (heavy and medium chain) intensity in western blots relative to β-actin (loading control). ****p < 0.0001 versus mock infection, n = 3. **(E)** Primary DRG neurons from WT and SARM1$^{-/-}$ mice cultured in microfluidic devices and infected with Thai dog rabies in the axon panel for 24 hours. The neurons are stained with pan axonal neurofilament antibody (green), anti α-tubulin (red) and anti-rabies nucleoprotein antibody (magenta). Scale bar 100 µM (maximum intensity projection of Z-stacks). Representative images show intact axons in SARM1$^{-/-}$ neurons but not in the WT DRG neurons infected with rabies. **(F)** Quantification of axonal degeneration as a ratio of disintegrated axons versus healthy (filamentous) (Axon degeneration index). ****p < 0.0001 versus mock, n = 4 (5 representative images quantified from each repeat). **(G)** Quantification of neurofilament immunostaining intensity in the axon panel images from experiments shown in Fig 3E & S5C Fig. ****p < 0.0001 versus mock infection, n = 4.

ratio between non-inoculated and inoculated panels for SARM1 knockout neurons, compared with wildtype neurons (Fig 4D). These data demonstrate that SARM1 mediated axonal degeneration induced in response to rabies infection impedes the axonal transmission of rabies virus between interconnected neurons.

## Discussion

This study identifies a novel role of SARM1/NAD mediated axonal and dendritic degeneration in rabies infection. Our results show that rabies infection triggers axonal degeneration mediated by NAD loss and calpain activation in both DRG and cortical neurons. This study further illustrates that SARM1 is required for the accelerated execution of rabies induced axonal degeneration. This intrinsic defense response impedes the axonal transmission of rabies virus but results in morphological damage to the neuronal network in ex-vivo models.

While SARM1 has been identified as a chief executor of axonal self-destruction, the mechanisms acting upstream of SARM1 activation remains unclear, although a few key regulators have recently been implicated. These include mitogen-activated protein kinase (MAPK) signaling [37] and Death Receptor 6 (DR6) [38] in mammals and Axundead in Drosophila [39]. In this study, deleting SARM1 gene significantly delayed axonal self-destruction in rabies infected ex-vivo neurons. However the degeneration of axons continued in the absence of SARM1, albeit at a much slower rate extending over several days. This implies that SARM1 acts as a rapid facilitator of axonal degeneration in viral infection by causing accelerated digestion of NAD molecules, similar to its role in axonal injury. However, it is still to be determined whether upstream molecules triggering the activation of SARM1 in viral infection are the same as those activating axonal self-destruction in injury (axotomy) or trophic withdrawal. Our identification of rabies induced axonal degeneration provides an alternate model to discover the activating mechanisms upstream of SARM1.

This study describes a new anti-viral defense mechanism mediated by SARM1 in response to rabies infection. This adds on to previously established roles of SARM1 in innate immune response such as negative regulation of toll-like receptor (TLR3) signaling [40] and induction of neuronal apoptosis in response to viral infection [41]. SARM1 is also shown to regulate intrinsic neuronal chemokine and cytokine response in traumatic axonal injuries [42]. The functional relationship between these multifaceted roles of SARM1 in innate immunity remains to be investigated.

The key pathological mechanisms associated with neurological dysfunction in rabies infection are yet to be fully defined. The paralytic form of rabies infection is linked to axonal neuropathy, muscle denervation, impaired nerve conduction and spinal paralysis [23, 43, 44]. In addition, features of axonal self-destruction such as focal axonal swellings, vacuolation, segmental demyelination and loss of nerve fibers are observed in the peripheral nerves of human

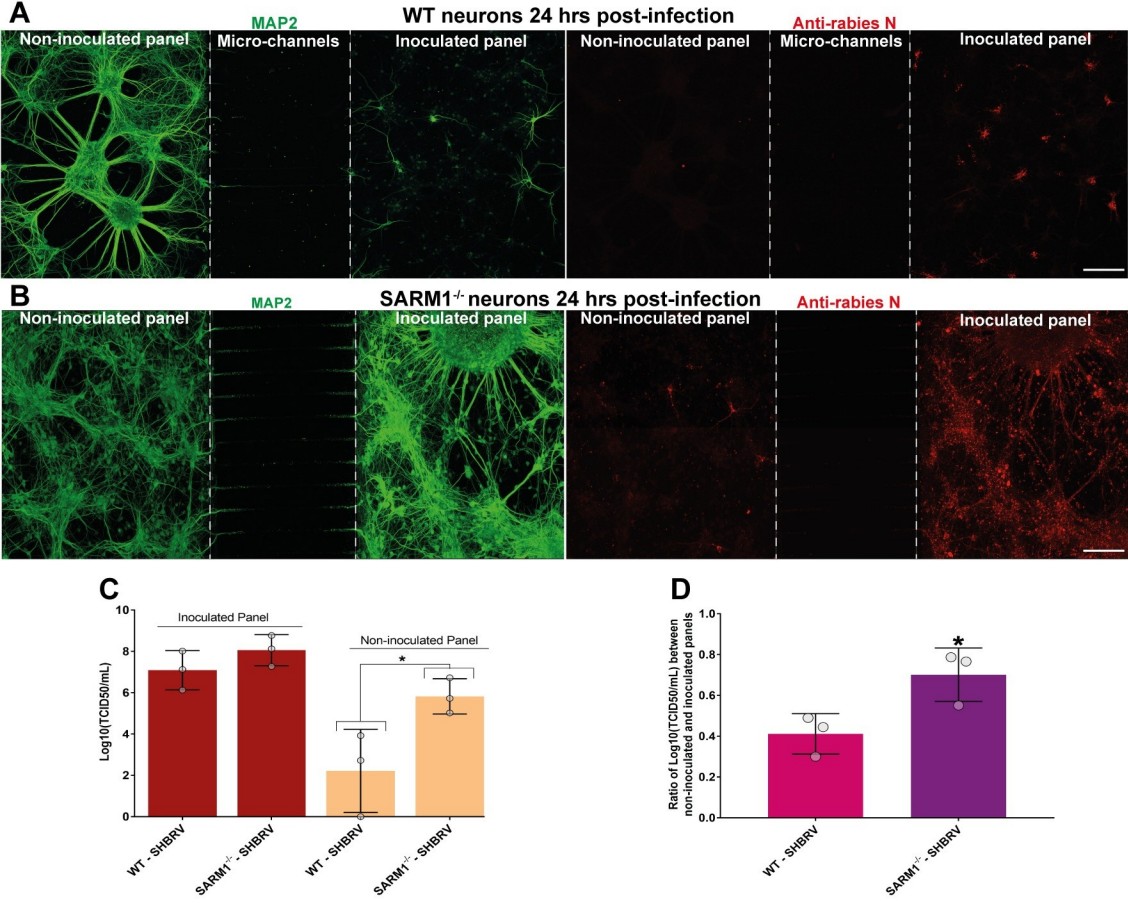

**Fig 4. Axonal self-destruction impedes the spread of rabies virus among synaptically connected neurons. (A)** Transynaptic transmission of rabies virus in WT neurons. Stitched tile confocal image (1.76 mm x 606.77 µM with maximum intensity projection of Z-stacks, 40 slices: 39.39 µM) showing the microfluidic chamber cultured with WT cortical neurons on both the panels (left side: non-inoculated panel and right side: inoculated panel). Neurons were infected with SHBRV Lyssavirus for 24 hours and a unidirectional flow of media from the non-inoculated panel into the inoculated panel, avoiding random diffusion of virus. After infection, neurons are stained with anti-MAP2 antibody (green) and anti-rabies nucleoprotein antibody (red). Neurons on the inoculated panel show significant loss of MAP2 immunostaining indicative of axonal self-destruction, whereas neurons in the non-inoculated panel show restricted rabies infection and normal filamentous MAP2 immunostaining. Scale bar 200 µM. **(B)** Transynaptic transmission of rabies virus in SARM1$^{-/-}$ neurons. Stitched tile confocal image (1.76 mm x 606.77 µM with maximum intensity projection of Z-stacks, 40 slices: 34.18 µM) showing the microfluidic chamber cultured with SARM1$^{-/-}$ cortical neurons on both the panels. Unlike the WT neurons, SARM1$^{-/-}$ neurons in the infected panel show normal filamentous MAP2 immunostaining, indicating lack of axonal self-destruction and efficient spread of rabies virus to the non-inoculated panel. Scale bar 200 µM. **(C)** Quantification of infectious rabies virus by titration assays in the culture media collected from the inoculated and non-inoculated panels of WT and SARM1$^{-/-}$ neurons. **(D)** Ratio of viral titre in the culture media from non-inoculated and inoculated panels of WT and SARM1$^{-/-}$ neurons. Data shows increased virus spread to the non-inoculated panel in the chambers cultured with SARM1$^{-/-}$ neurons, as compared to WT neurons. $^*$p < 0.05; SARM1$^{-/-}$ versus WT neurons infected with SHBRV rabies, n = 3.

patients with the paralytic form of rabies [21, 23, 25]. This peripheral nerve damage also correlates with reduced spread of virus to the brain in the paralytic form, in contrast to the furious form of rabies [24, 45, 46]. Thus, the pathological features of paralytic rabies are strongly in agreement with our findings in ex-vivo neuron cultures, including axonal self-destruction and impediment of viral spread. Our study suggest that SARM1/NAD mediated axonal self-destruction activated by neurons in response to rabies infection, could result in peripheral nerve damage and neurological dysfunction in the paralytic form of rabies.

Interestingly, the degenerative axonal pathology in the peripheral nervous system is observed mainly in the paralytic form of rabies but not in the furious form [23]. Furious rabies

is likely to be associated with dysfunction in the brain [47], as it affects higher neurological functions such as behavior and cognition [48]. Furthermore, there is usually minimal spinal paralysis in the furious form, at least in the initial stages of disease. This suggests a lack of or delayed axonal self-destruction in the furious form, allowing unhindered passage of virus to the brain. As observed in the cortical neurons in our study, axonal and dendritic degeneration could be induced by rabies infection in the brain at later stages of infection in the furious form. Indeed, by depleting SARM1 expression we show that axonal degeneration is delayed and degeneration occurs much later, allowing sufficient time for the virus to spread through the axons. However, it is unclear how in the furious form, which constitutes the majority of rabies infections, the virus is able to overcome SARM1 mediated axonal self-destruction in peripheral nerves and is able to spread through the nervous system. Many factors including differences in host-adaption of viral strains and the pre-existing immune status of the human host may contribute to the lack of peripheral axonal degeneration in the furious form of rabies. Improved animal models representing the two clinical forms of human rabies infection are therefore required in future studies, to conclusively validate the role of axonal self-destruction in determining rabies clinical outcome.

In summary, our study identifies that axonal self-destruction may be activated by neurons in response to viral infection. To our knowledge, this presents the first identification of the role of SARM1 pathway in evoking axonal degeneration in rabies infection. This describes a novel signaling mechanism responsible for dysfunction of neurons in rabies infection. These results therefore warrant further examination of axonal self-destruction in viral infections other than rabies, which involve axonal trafficking of virions. This study significantly improves our understanding of intrinsic neuronal response to viral infection and opens the door to identification of new avenues for the treatment of neurotropic viral infections.

## Materials and methods

### Ethics statement

All experiments involving mouse embryos in this study were reviewed and approved by the Animal Ethics Committee (AEC) of the Australian Animal Health Laboratory (AAHL) (approval #1880, 1900 and 1901), following the Australian National Health and Medical Research Council Code of Practice for the Care and Use of Animals for Scientific Purposes.

### Mice

Wildtype C57BL/6J mice were purchased from Animal Resource Centre (Western Australia) and housed at the Small Animal Facility (SAF) of AAHL. SARM1 knockout mice on the C57BL/6 background developed previously [36], were purchased from Jackson Laboratories (USA) (stock number 018069, RRID:IMSR_JAX:018069). SARM1 knockout mice were housed and time-mated at SAF-AAHL in the same conditions as wildtype mice. Genotyping of SARM1$^{-/-}$ mice was performed following the standard PCR protocol provided by the Jackson Laboratories, using the following primer sequences (5' to 3'); 23149: GGG AGAGCCTTCCT CATACC, 23150: TAAGGATGAACAGGGCCAAG, oIMR6916: CTTGGG TGGAGAGGC TATTC and oIMR6917: AGGTGAGATGACAGGAGATC.

### Viruses

Original lyssavirus strains isolated from infected dog or bat were amplified in suckling mouse brain (Swiss mice) by intracerebral inoculation. The brain homogenates containing the virus were then used to infect Neuro-2a cells for further amplification. The viral strains were

subjected to fewer than 3 passages in Neuro-2a cells during amplification. For infection with parental viral stocks, brain homogenates prepared from infected dog brain (P0) or after first passaging in sucking mouse brain (P1) were used. The brain homogenates were clarified and concentrated using Amicon 10kDa Ultra centrifugal filters (Merck). The viral titre of cell culture supernatants and clarified brain homogenates were determined by titration assays on BHK-21 cells. All experiments involving viruses were performed in the biosafety level 3 (BSL3) laboratories at AAHL, following protocols approved by AAHL's institutional biosafety committee.

## Method details

**Primary neuron cultures.** Primary cortical neuron cultures were generated from embryonic day 15 (E15) embryos from both wildtype and SARM1$^{-/-}$ mice, based on previously published protocols [49]. Briefly, cortices from E15 embryos were separated under aseptic conditions, chopped into small pieces and digested with 0.125 mg/mL trypsin (Sigma-Aldrich) for 15–20 mins at 37˚C. The tissues were then treated with soybean trypsin inhibitor (STBI; Sigma-Aldrich) and DNase I (8000 units; ThermoFisher) and homogenised gently to form uniform cell suspensions. Cortical neurons were seeded at 120,000 cells per well on poly-L-ornithine (Sigma-Aldrich) coated glass coverslips (13mm; Menzel Glaser) in 24 well plates. Primary DRG neurons were generated from E13 or E14 embryos. The DRGs were isolated from the embryos and subjected to trypsin digestion and homogenisation as above. Both cortical and DRG neurons were cultured in neurobasal media with B27 supplement, glutamax (ThermoFisher) and gentamicin (Sigma-Aldrich).

**Neuronal culture in microfluidic chamber.** Compartmentalized axon chambers: For separating the neuronal cell bodies and axons, cortical and DRG neurons were cultured in xona microfluidic devices [28] (XONA Microfluidics, Cat#SND450) mounted on glass coverslips (24 x 40 mm; Menzel Glaser) coated with poly-L-ornithine (Sigma-Aldrich). Approximately 10 μL of cell suspension containing 120,000 cells were added to the cell body panel. The chambers were then incubated at 37˚C for 10 min to allow attachment of neurons. Then 200 μL of neuronal culture media was added to the top and bottom wells of cell body panel and 150 μL of media was added to the wells in the axon panel. Approximately half the volume of media in each well was replaced with fresh media every two days and the higher volume of media on the cell body panel was maintained.

Transynaptic microfluidic model: To model synaptically connected ex-vivo neuronal cultures, cortical neurons were extracted as above and seeded on to both the panels of microfluidic device (SND450, XONA microfluidics), each with 10 μL of cell suspension containing 120,000 cells.

**Viral quantification.** Viral titres of inocula were determined by direct fluorescent antibody test in BHK cells. Serial 10-fold dilutions of viral suspensions were prepared in cell culture media and were added to 96 well plates (4 replicates each), followed by BHK cell suspensions. The cells were then incubated at 37˚C with 5% $CO_2$ for 5–6 days. The plates were fixed with 10% formalin for 30 minutes at room temperature and then stained with FITC conjugated anti-rabies monoclonal antibody (Fujirebio) at 1:10 dilution in 0.5% BSA/PBSA with 0.005% Evans blue. Plates were read with an Olympus BX51 inverted microscope and the median tissue culture infectious dose (TCID$_{50}$) determined [50].

**Viral infection of primary neuron cultures.** Primary DRG and cortical neuron cultures were infected after 7–10 days of culture. For infection of neurons cultured in 24 well plates and microfluidic chambers, viral inoculum containing the titre required to infect the neurons at MOI of 1 (based on the number of neurons originally seeded in each well or panel in

microfluidic chamber, i.e. 120,000 cells) were added. For the microfluidic chambers media from the axon panel was removed and viral inoculum was added to the top well of the axon panel and allowed to flow through to the bottom well. Then the appropriate volume of media was added, so the total volume in each well in the axon panel was 150μL. Then 200 μL of media was added to the cell body panel, so a unidirectional flow of media from the cell body to axon panel was maintained. Similarly, to infect trans-synaptically connected neurons in microfluidic chambers, media from the panel to be inoculated was removed and the viral inoculum added. A unidirectional flow of media from the non-inoculated to inoculated panel was always maintained with higher volume of media in the non-inoculated panel. For mock infection, cell culture supernatant collected from uninfected Neuro-2a cells was added to the wells or chambers.

**Drug treatments.** Drug treatments were performed on infected primary cortical neurons cultured on coverslips in 24 well plates. Media from the wells was removed and replaced with fresh media with the appropriate volume of viral inoculum to infect the neurons at MOI of 1. After adding the viral inoculum containing media, appropriate dilutions of NAD (Sigma-Aldrich), EGTA (Sigma-aldrich) or calpain inhibitor III (ABCAM) were added to each well. The stock solution of these compounds was prepared as follows: NAD– 50mM and 200mM, dissolved in sterile water; EGTA– 200mM dissolved in 0.2M NaOH; Calpain inhibitor III - 20mM dissolved in DMSO. The neurons were fixed at 24 hours post-infection and analysed by immunostaining and confocal imaging.

**Immunocytochemistry.** Primary cortical neurons cultured on coverslips were fixed with 4% paraformaldehyde (PFA, Sigma-Aldrich) in 0.05 M phosphate buffered saline (PBSA) for 1hour at room temperature. The coverslips were washed gently three times with PBSA and the cells were permeabilized with 0.1% Triton X-100 (Sigma-Aldrich) in PBSA for 5 min and then rinsed with PBSA. They were then blocked with 0.5% BSA in PBSA for 30 min and incubated overnight at 4˚C with primary antibodies diluted in 0.5% BSA in PBSA. They were washed three times with PBSA and incubated with species-specific fluorescent secondary antibodies (Alexa Fluor, ThermoFisher) diluted at 1:200 in 0.5% BSA in PBSA for 1 hour at room temperature. Coverslips were then rinsed twice with PBSA, twice with sterile water and then stained with DAPI for 10 min, then were washed twice with sterile water and mounted on glass slides (ThermoFisher) with Vectashield mounting medium (Vector Laboratories).

The following primary antibodies were used at the indicated dilutions: chicken anti-MAP2 (1:1000, ABCAM, cat#ab4674), rabbit anti-rabies nucleoprotein, 1:3000, in-house [51], mouse pan-axonal neurofilament antibody (SMI-312, 1:1000, BioLegend, cat#837904), Mouse anti- α tubulin (1:1000, Sigma-Aldrich, cat#T6199) and chicken anti- α tubulin (1:1000, ABCAM, cat#ab89984).

**Immunostaining of neurons in microfluidic chambers.** Primary cortical and DRG neurons cultured in microfluidic chambers were fixed with 4% PFA after infection. 200μL of PFA was added to all four wells in the chamber and allowed to fix for at least an hour. After fixation, immunostaining with primary and secondary antibodies were performed in the microfluidic chamber as above, without disturbing the attachment to glass coverslips. For washing, PBSA was added to the top wells first, was allowed to flow through to the bottom wells, was incubated for 10 mins and then repeated three times. Primary and secondary antibodies were appropriately diluted (as above) in 0.5% BSA in PBSA and 200μL was added to each well of the chamber during incubation. After final washes, PBSA was added to all the wells in the chamber and stored at 4˚C before confocal imaging.

**Confocal and live-cell imaging.** Confocal imaging was performed using a ZEISS LSM 800 inverted confocal microscope. To analyse neurite degeneration in MAP2-stained cortical neurons, images were acquired with the 20x objective using tiling function, covering an area of 998.28 x 673.84μM with Z-stacks (20 slices: 19–24μM), including at least 500 neurons in each

image. The images were then stitched and a maximum intensity projection of z-stacks were generated. All the confocal imaging and processing were performed using ZEN 2.5 Blue software (ZEISS).

For imaging the axon panels of DRG neurons, immunostained microfluidic chambers were placed carefully on the inverted microscope stage. Images of the axon panels were then taken with a 20x objective using tiling function, covering an area of 607.08 x 603.34µM with Z-stacks (40 slices: 14–18µM). Similarly, for imaging trans-synaptic neuronal cultures in microfluidic chamber, tile images were taken covering an area of 1.76 mm x 606.77µM with Z-stacks (40 slices: 34–40µM).

Live-cell DIC imaging of rabies infected cortical and DRG neurons were performed using Leica SP5 confocal microscope in BSL3. Neurons were cultured in glass bottom dishes (µ-Dish 35 mm high glass bottom, Ibidi), infected with SHBRV lyssavirus at MOI-1 and imaged for 24 hours after infection.

**Image analysis of axonal and dendrite degeneration.**   Quantification of MAP2-positive dendrites relative to DAPI stained nuclei was performed using stitched tile confocal images, using ImageJ [52]. Images were binarized, threshold normalized to mock-infected neuron images and the integrated density of MAP2 immunofluorescence was measured. This value was then divided by the total number of DAPI stained nuclei in the same image, counted using particle analyzer plugin with the following parameters: size (inch$^2$)– 0.005-infinity and circularity– 0.00–1.00. "Watershed" function was applied to the DAPI stained images before analysis to avoid counting merged particles.

The axon degeneration index from tubulin stained confocal images of axons in microfluidic chambers was calculated based on a method described previously [53]. The tubulin stained axon images were binarized and the total axon area was measured. Then the fragmented axons were quantified using particle analyzer plugin with the following parameters: size (inch$^2$)– 20–15000 and circularity– 0.00–1.00. Axon degeneration index was then calculated as the ratio of fragmented axon area over total axon area. To quantify axonal degeneration by neurofilament loss in DRG neurons, the integrated density of neurofilament immunofluorescence in DRG axons were quantified from binarized images after normalizing threshold to mock-infected neuron images.

**Apoptosis assay.**   To quantify apoptotic cells in rabies infection, TUNEL staining was performed on the PFA fixed neurons on coverslips, according to manufacturer's protocol using the in-situ cell death detection kit, TMR red (Sigma-Aldrich, cat#12 156 792 910). As a positive control for apoptosis, fixed and permeabilised neurons were treated with 3 units of DNase I in 50 mM Tris-HCL for 15 mins to induce DNA strand breaks. Confocal images of neurons stained with TUNEL TMR red and DAPI were taken as tile images covering an area of 998.28 x 673.84 µM with Z-stacks (20 slices: 19–24 µM), including at least 500 neurons in each image. The percentage of TUNEL-positive apoptotic bodies in these images (threshold normalized to controls), relative to DAPI stained nuclei were counted using imageJ particle analyzer plugin and watershed function, with the following parameters: size (inch$^2$)– 0.003-infinity and circularity– 0.00–1.00.

**Western blotting.**   For western blotting, cortical neurons were lysed with RIPA buffer (Sigma-Aldrich) with 0.1% SDS and protease inhibitor cocktail (Sigma-Aldrich). The protein lysates were heated at 56˚C for 30 mins to inactivate rabies virus [54]. The protein concentration in the lysates were determined using DC protein assay kit (Bio-Rad). 30µg of lysates were mixed with NuPAGE LDS sample buffer (ThermoFisher), heated at 90˚C for 5 mins and then loaded on to Bolt 4–12% Bis-Tris Plus Gels (ThermoFisher). The gels were run at 150 volts for 1hour in Bolt MOPS SDS running buffer (ThermoFisher). Proteins were then transferred onto PVDF membrane blots (Bio-Rad) using wet tank blotting system (Bio-Rad), blocked with 3%

BSA in PBSA with 0.2% Tween 20 (Merck Millipore) and probed overnight at 4˚C with the following concentrations of primary antibodies diluted in 1% BSA/PBSA: chicken anti-MAP2 (1:1000, ABCAM, cat#ab92434), mouse pan-axonal neurofilament antibody (1:1000, BioLegend, cat#837904) and mouse anti- β actin (1:2000, Sigma-Aldrich, cat#A2228). The blots were then washed with 0.5% Tween 20 in PBSA and incubated with appropriate species-specific fluorescent secondary antibodies (Alexa Fluor, ThermoFisher) for 1hour at room temperature. The blots were imaged using iBright FL1000 imaging system (ThermoFisher).

**Statistical analysis.** Statistical significance between two values was determined using a two-tailed t test, analyses of three or more values were performed by one-way ANOVA with Bonferroni's post hoc test using Prism 7 (GraphPad).

Data are presented as mean ± SEM. $^*p<0.05$, $^{**}p<0.01$, $^{***}p<0.001$, $^{****}p<0.0001$.

## Supporting information

**S1 Fig. Effect of rabies infection on primary cortical neurons. (A)** Rabies induced blebbing of MAP2-positive neurites in cortical neurons at 16 hours post infection (high magnification). Representative confocal images of mouse primary cortical neurons infected with rabies strains (H.ABLV, SHBRV and Z.Dog), showing focal swellings in the MAP2-positive neurites (white arrows) at 16 hours post infection. Neurons are stained with anti-MAP2 antibody (green) and anti-rabies nucleoprotein antibody (Red). Scale bar 10 μM**. (B)** Rabies induced loss of MAP2--positive neurites in cortical neurons at 24 hours post infection (low magnification). Representative stitched tile images of mouse cortical neurons infected with rabies strains (H.ABLV, SHBRV and Z.Dog) showing loss of MAP2 staining at 24 hours post infection, but not in mock infected neurons. Scale bar 100 μM. Neurons are stained with anti-MAP2 antibody (green), anti-rabies nucleoprotein antibody (red) and DAPI (blue). Merged images of all three channels are shown.
(TIF)

**S2 Fig. Degeneration of axons and dendrites in rabies infected neurons. (A)** Representative confocal images of mouse primary cortical neurons infected with parental stocks of rabies strains (Z.Dog P1 and T.Dog P0). High magnification images of mock infected neurons show co-staining of shorter dendrites with MAP2 antibody (green) and longer axons with neurofilament antibody (red) in the same neurons. Infection with Z.Dog P1 and T.Dog P0 strains for 24 hours, show significant loss of filamentous dendritic and axonal staining in neurons. Rabies infection is identified by staining with anti-rabies nucleoprotein antibody (magenta) and nuclei are stained in blue. Scale bar 20 μM. **(B)** Examination of induction of apoptosis in primary cortical neurons infected with rabies for 24 hours. Representative stitched tile images of mouse cortical neurons infected with rabies strains and stained with TUNEL to detect apoptosis. As a positive control, neurons were treated with DNase I to induce apoptotic DNA fragmentation. Images show neurons stained with DAPI (blue-nuclei) and TUNEL (red-apoptotic neurons). Scale bar 100 μM. **(C)** Quantification of apoptotic neurons in rabies infected and mock infected cultures. Percentage of apoptotic neurons was calculated from experiments shown in S2B Fig using ImageJ. TUNEL positive neurons were identified as apoptotic and the percentage of apoptosis was calculated from the total number of DAPI stained nuclei. At least 500 neurons are quantified per sample. Data was found to be not significant versus mock infected neurons, n = 4. Images shown are maximum intensity projections of Z-stacks.
(TIF)

**S3 Fig. Axonal infection of primary cortical neurons cultured in microfluidic chambers with rabies virus. (A)** Immunostaining of mock infected primary cortical neurons cultured in

microfluidic chamber. Stitched tile confocal image showing the microfluidic chamber seeded with cortical neurons on the cell body panel. As the neurons grow, axons extend through the micro-channels into the axon panel. The neurons were mock infected on the axon panel for 24 hours and a higher volume of culture media was maintained on the cell body panel to maintain the flow of media from cell body to axon panel and avoid random diffusion of inoculum from the axon panel. Neurons are stained with anti-neurofilament antibody (NF, green), anti α-tubulin (red) and anti-rabies nucleoprotein antibody (magenta). Scale bar 100 μM. **(B)** Higher magnification images of the axon panel from the mock infected neurons shown in S3A Fig. Images show intact neurofilament staining and filamentous tubulin stained axons. Scale bar 50 μM. **(C)** Immunostaining of primary cortical neurons cultured in microfluidic chamber and infected with Z.Dog rabies virus for 24 hours in the axon panel. Stitched tile confocal image show loss of neurofilament staining in the axons of neurons infected with rabies for 24 hours, while neurofilament staining can still be observed in the cell bodies and the proximal axons in the cell body panel. Similarly, tubulin staining in the axon panel show granular disintegration of distal axons. Rabies infection is identified by positive staining with anti-rabies nucleoprotein antibody. Scale bar 100 μM. **(D)** Higher magnification images of axon panel from the Z.Dog rabies infected neurons shown in S3C Fig. Images show axonal degeneration as observed by the loss of neurofilament staining and granular disintegration of tubulin stained axons. Scale bar 50 μM. Images shown are maximum intensity projections of Z-stacks.
(TIF)

**S4 Fig. (A) Rabies induced axonal degeneration in primary DRG neurons cultured in microfluidic chambers.** Mouse DRG neurons were cultured in microfluidic devices to separate the cell body and axons. The axon panels were then infected with different street strain rabies virus (H.ABLV, SHBRV and Z.Dog) for 24 hours. Figure shows stitched tile images of axon panel infected with rabies virus. The axons were stained with pan axonal neurofilament antibody (green), anti-rabies nucleoprotein antibody (magenta) and anti α-tubulin (red). Scale bar 100 μM. Axonal degeneration is observed by the loss of neurofilament staining and granular disintegration of tubulin stained axons. Images shown are maximum intensity projections of Z-stacks. **(B) Pharmacological inhibition of neurite degeneration in rabies infection.** Effect of extracellular NAD addition and calpain inhibition on rabies induced loss of MAP2-positive neurites. Stitched tile images of mouse cortical neurons infected with H.ABLV, SHBRV and Z.Dog rabies for 24 hours and treated with NAD, EGTA or Calpain inhibitor III. Scale bar 100 μM. Neurons are stained with anti-MAP2 antibody (green), DAPI (blue) and anti-rabies nucleoprotein antibody (red). All three channels are merged together, which show reduced loss of MAP2-positive neurites in infected neurons treated with NAD, EGTA or calpain inhibitor III compared to no drug treatment. Images shown are maximum intensity projections of Z-stacks.
(TIF)

**S5 Fig. Deletion of SARM1 gene significantly delays axonal degeneration in primary cortical and DRG neurons infected with rabies. (A)** Time course analysis of neurite degeneration in SARM1 knockout neurons infected with rabies. Representative images of mouse cortical neurons from SARM1 knockout (SARM1$^{-/-}$) mice infected with rabies virus (H.ABLV, SHBRV and Z.DOG) for 24 hours to 7 days. Images show intact MAP2-positive neurites in rabies infected SARM1$^{-/-}$ neurons at 24 hours, but a gradual loss of MAP2-positive neurites extending up to 7 days. Neurons are stained with anti-MAP2 antibody (green), anti-rabies nucleoprotein antibody (red) and DAPI (blue). Scale bar 100 μM. **(B)** Immunostaining of SARM1$^{-/-}$ cortical neurons cultured in microfluidic chamber and infected on the axonal panel with Z.Dog rabies virus for 24 hours. Stitched tile confocal image shows SARM1$^{-/-}$ neurons

infected on the axon panel for 24 hours. Neurons are stained with anti-neurofilament antibody (NF, green), anti α-tubulin (red) and anti-rabies nucleoprotein antibody (magenta). Scale bar 100 μM. Image shows lack of axonal degeneration in SARM1$^{-/-}$ neurons infected specifically at the axons with rabies virus. **(C)** Inhibition of rabies induced axonal degeneration in SARM1$^{-/-}$ DRG neurons cultured in microfluidic chambers. Primary DRG neurons from WT and SARM1$^{-/-}$ mice cultured in microfluidic devices to separate the cell body and axons. The axon panels were then infected with different street strains of rabies virus (H.ABLV, SHBRV and Z. DOG) for 24 hours. Figure shows stitched tile images of axon panel of SARM1$^{-/-}$ neurons infected with rabies. The axons were stained with anti-neurofilament antibody (green), anti α-tubulin (red) and anti-rabies nucleoprotein antibody (magenta). Scale bar 100 μM. Images shown are maximum intensity projections of Z-stacks.
(TIF)

**S1 Video. Live DIC imaging of primary cortical neurons infected with SHBRV Lyssavirus.** Infected neurons were imaged from 2 hours up to 24 hours post-infection. Video shows axonal swellings, granular disintegration and loss of axons. Axons were imaged every 10 minutes. Scale bar 50 μM.
(AVI)

**S2 Video. Live DIC imaging of primary DRG neurons infected with SHBRV Lyssavirus.** Infected neurons were imaged from 2 hours up to 22 hours post-infection. Video shows granular disintegration and loss of axons. Axons were imaged every 15 minutes. Scale bar 25 μM.
(AVI)

## Acknowledgments

We are grateful for the support of Pathology & Pathogenesis, Small Animal Facility and Tissue Culture teams at AAHL. The Bioassays R&D Team provided the anti-rabies nucleoprotein antibody used in this study. We thank Dr. Christine Fehlner-Gardiner, Centre of Expertise for Rabies, Canadian Food Inspection Agency and the Thailand Department of Livestock Development (DLD) and National Institute of Animal Health (NIAH) for providing rabies viral isolates. We thank Gemma Carlile and Anthony Keyburn for organising importation of rabies isolates from Thailand. We also thank Nathan Godde and Ryan Farr for stimulating intellectual discussions. The Australian Microscopy & Microanalysis Research Facility (AMMRF) supported the confocal and live-cell imaging capability utilised in this study.

## Author Contributions

**Conceptualization:** Vinod Sundaramoorthy, Carmel M. O'Brien, Megan Dearnley, John Bingham.

**Formal analysis:** Vinod Sundaramoorthy.

**Investigation:** Vinod Sundaramoorthy, Diane Green, Kelly Locke.

**Methodology:** Vinod Sundaramoorthy.

**Project administration:** Megan Dearnley, John Bingham.

**Supervision:** Megan Dearnley, John Bingham.

**Visualization:** Vinod Sundaramoorthy.

**Writing – original draft:** Vinod Sundaramoorthy.

Writing – review & editing: Carmel M. O'Brien, Megan Dearnley, John Bingham.

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
