## [Decision Letter · Decision Letter 0]

13 Nov 2019

Dear Dr Sundaramoorthy,

Thank you very much for submitting your manuscript "Novel role of SARM1 mediated axonal degeneration in the pathogenesis of rabies." (PPATHOGENS-D-19-01797) for review by PLOS Pathogens. Your manuscript was fully evaluated at the editorial level and by independent peer reviewers. The reviewers appreciated the attention to an important problem, but raised some substantial concerns about the manuscript as it currently stands. These issues must be addressed before we would be willing to consider a revised version of your study. We cannot, of course, promise publication at that time.

All three reviewers thought this was an excellent study appropriate for a broad audience, and, if some of the major concerns could be addressed, would meet the standard of advance we require at PLoS Pathog. However, the first two reviewers raised concerns, which need to be addressed experimentally. Reviewer #1 raised an important point that unequivocal discrimination between axons and dendrites is critical for the authors' underlying mechanistic interpretation; as is stands, the MAP2 staining alone is insufficient to make this discrimination for the reasons stated by the reviewer. Reviewer #2 had some concerns with the utilized RABV strain - you should consider confirming some of your results with another RABV strain to ensure it's not specific to the one used RABV strain. Many of Reviewer #2's other concerns could also be addressed by further explication or toning down the conclusions.   

We therefore ask you to modify the manuscript according to the review recommendations before we can consider your manuscript for acceptance. Your revisions should address the specific points made by each reviewer.

(1) A letter containing a detailed list of your responses to the review comments and a description of the changes you have made in the manuscript. Please note while forming your response, if your article is accepted, you may have the opportunity to make the peer review history publicly available. The record will include editor decision letters (with reviews) and your responses to reviewer comments. If eligible, we will contact you to opt in or out.

(2) Two versions of the manuscript: one with either highlights or tracked changes denoting where the text has been changed; the other a clean version (uploaded as the manuscript file).

Additionally, to enhance the reproducibility of your results, PLOS recommends that you deposit your laboratory protocols in protocols.io, where a protocol can be assigned its own identifier (DOI) such that it can be cited independently in the future. For instructions see http://journals.plos.org/plospathogens/s/submission-guidelines#loc-materials-and-methods

We hope to receive your revised manuscript within 60 days. If you anticipate any delay in its return, we ask that you let us know the expected resubmission date by replying to this email. Revised manuscripts received beyond 60 days may require evaluation and peer review similar to that applied to newly submitted manuscripts.

[LINK]

Sincerely,

Matthias Johannes Schnell, PhD

Associate Editor

PLOS Pathogens

Benhur Lee

Section Editor

PLOS Pathogens

Kasturi Haldar

Editor-in-Chief

PLOS Pathogens

orcid.org/0000-0001-5065-158X

Grant McFadden

Editor-in-Chief

PLOS Pathogens

orcid.org/0000-0002-2556-3526

Reviewer's Responses to Questions

**Part I - Summary**

Reviewer #1: In this paper, Sundaramoorthy et al., show that cultures of primary cortical and peripheral neurons infected with different lyssavirus strains (rabies virus, RABV) lose structural proteins and demonstrate neurite degeneration. This is a striking phenotype resulting in near complete loss of neurite structures by 24 hours after infection, supported by MAP2, NF and tubulin antibody staining. They also show that SARM1 knockout mouse neurons, when infected with RABV, show delayed neurite degeneration with subsequent enhanced transneuronal RABV spread in cortical neurons cultured in connected microfluidic chambers. This is an important finding hinting at an intrinsic neuronal defense mechanism against viral spread possibly leading to RABV related neuropathologies. Although, the paper was written clearly and results were supported by clean images and quantitation, some critical data was missing.

One of the critical points missing is the discrimination between axons and dendrites. The authors claim in many experiments that the neurites referred to are axons (particularly in the last figure), however, only MAP2 staining is shown. MAP is abundant in somatodendritic regions, but not axons. This point is important because RABV is known to spread retrogradely between connected neurons after entering axons of motor neurons from neuromuscular junctions. That fact means that the endocytosed particles are transported in the axons, the genomes are released and replicate in the neuronal soma, and progeny spread from dendrites to connected axons. If the infected cell axon degenerates, it will not reduce the somatodendritic spread in the infected neuron. If all neurites (both axons and dendrites) are degenerating due to a loss of MAP2, then SARM1 mediated axon degeneration is not the predominant mechanism blocking transneuronal viral spread. As an alternative hypothesis, recall that Sarm1 was originally identified as a negative regulator of TLR3 and TLR4 pathways in innate immunity (Carty et al., 2006, Nat. Immunol.). This negative regulation might be responsible for the virus yield phenotype in Sarm1 knock out neurons. This alternative hypothesis is important particularly for interpreting the last figure (F4). It appears that there is more viral antigen staining in the inoculated chamber of SARM1 knock out neurons. However, in figure 3, it appears that there is less viral antibody staining 24 hours after infection, but staining catches up after 4 days. Virus yields should be compared at high MOI between wild type and knock out conditions.

In general, the authors have a solid observation of the loss of cytoskeletal proteins after RABV infection, but the role of SARM1 might not be limited to the delayed spread (due to delayed axon degeneration). Instead, SARM1 effects might involve innate immunity as well. More detailed characterization of axons vs dendrites is required to make that distinction.

Reviewer #2: The paper from Vinod Sundaramoorthy et al entitled « Novel role of SARM1 mediated axonal degeneration in the pathogenesis of rabies” presents data to describe a potential Rabies virus (RABV) mediated axonal self-destruction mechanism which could account for RABV pathology. The experimental approach is based on in vitro neuronal cultures, imaging techniques and Xona microfluidic devices.

Reviewer #3: Using rabies virus field isolates, this report shows that, in response to infection, neurons activate selective degeneration of processes, mediated by the loss of NAD and digestion of cytostructural proteins. The authors then show that SARM1, known to be associated with axonal degeneration, is key to this process, likely restricting the ability of the virus to spread trans-synaptically. This is a beautifully done study: the images are compelling, the data are definitive, and the “story” is simple but powerful. Two minor comments:

1. It is difficult to ascertain, even from the high res file, but it appears that the nuclei in infected mice in Figure 1B are smaller and more fragmented than in the mock control. This is also seen in other images presented, but not always consistently. Is this the case? If samples are collected later, do the nuclei remain apparent, or is this a “stepwise” process in which neurites are lost followed by neuronal loss? TUNEL may not capture other forms of cell death.

2. This is, exclusively, an ex vivo study using primary neurons. While I do not believe it is compulsory to perform experiments in mice, I am curious if RV-infected, SARM KO mice differ from wild type mice in terms of pathogenesis. Addition of in vivo data (if it is even possible to obtain) would add merit to the significance of the work.

**Part II – Major Issues: Key Experiments Required for Acceptance**

Reviewer #1: The identification of neuritis as axons is critical. The MAP2 staining is not convince proof that they are imaging axons.

The alternative hypothesis that the effect of SARM1 is to negatively regulate TLR3 and 4 pathways must be considered/discussed.

Virus yields should be compared at high MOI between wild type and SARM1 knock out infections.

Reviewer #2: Although this work is extremely interesting and paves the way for further understanding how RABV can kill its host, I have several concerns.

The choice of the virus strains: The authors claimed they have chosen field strain lyssaviruses in order to limit or avoid any artefact regarding the original pathogenicity. This is a very good statement but, as described in the material and methods, the viruses have been amplified in suckling mouse brain and passaged in Neuro-2a cells (<3 passages). The viruses are then mouse adapted. Therefore, the authors should provide NGS sequencing of the field isolates before, during and after selection and passages in order to show the consensus sequence and the viral population diversity.

On the other hands and on the contrary of the authors, I think a highly virulent laboratory fixed strain, which exhibits Rabies encephalitis in mice, would be extremely informative and should be included in this study.

The choice of the cellular models: All the experiments are undertaken either with primary cortical neurons (embryonic day 15) and primary DRG neurons (embryonic day 13 or 14). Therefore, the study is carried out on immature neurons. Since neuron survival can be neuron type-specific and/or development stage-specific (see for example AJ Kole et al-2013, Ulrich Pfisterer et al-2017), the authors have to establish that their conclusions are still relevant in mature neurons.

Although, I would suggest they show at least some data in iPS derived human neurons in order to extend their findings to other species.

Statement: The authors claimed that: “this presents the first identification of a direct mechanism for rabies induced dysfunction of neurons”. It looks to me that is a bit overstated. The authors may want to cite other articles such as (not exclusive): Alan C. Jackson et al et al-2010, Jeison Monroy-Gomez et al-2018 and so on.

Reviewer #3: (No Response)

**Part III – Minor Issues: Editorial and Data Presentation Modifications**

Reviewer #1: (No Response)

Reviewer #2: (No Response)

Reviewer #3: (No Response)

PLOS authors have the option to publish the peer review history of their article (what does this mean?). If published, this will include your full peer review and any attached files.

Reviewer #1: No

Reviewer #2: No

Reviewer #3: No

---

## [Editor Report · Decision Letter 1]

22 Jan 2020

Dear Dr Sundaramoorthy,

We are pleased to inform you that your manuscript 'Novel role of SARM1 mediated axonal degeneration in the pathogenesis of rabies.' has been provisionally accepted for publication in PLOS Pathogens.

Before your manuscript can be formally accepted you will need to complete some formatting changes, which you will receive in a follow up email. A member of our team will be in touch within two working days with a set of requests.

Best regards,

Matthias Johannes Schnell, PhD

Associate Editor

PLOS Pathogens

Benhur Lee

Section Editor

PLOS Pathogens

Kasturi Haldar

Editor-in-Chief

PLOS Pathogens

orcid.org/0000-0001-5065-158X

Michael Malim

Editor-in-Chief

PLOS Pathogens

orcid.org/0000-0002-7699-2064

Editor's note: We acknowledge the authors' efforts to address the major criticisms of the reviewers in a judicious and constructive manner.  This enabled the final decision to be made on an editorial level.   
---

## [Editor Report · Acceptance letter]

7 Feb 2020

Dear Dr Sundaramoorthy,

We are delighted to inform you that your manuscript, "Novel role of SARM1 mediated axonal degeneration in the pathogenesis of rabies.," has been formally accepted for publication in PLOS Pathogens.

Best regards,

Kasturi Haldar

Editor-in-Chief

PLOS Pathogens

orcid.org/0000-0001-5065-158X

Michael Malim

Editor-in-Chief

PLOS Pathogens

orcid.org/0000-0002-7699-2064